# Variation and Genetic Control of the Heartwood, Sapwood, Bark, Wood Color Parameter, and Physical and Mechanical Properties of *Dipteryx panamensis* in Costa Rica

Dalina Rodríguez-Pérez, Róger Moya *, Olman Murillo, Johana Gaitán-Álvarez and Yorleny Badilla-Valverde

Escuela de Ingeniería Forestal, Instituto Tecnológico de Costa Rica, Apartado, Cartago 159-7050, Costa Rica; dalirp.08@gmail.com (D.R.-P.); omurillo@itcr.ac.cr (O.M.); jgaitan@itcr.ac.cr (J.G.-Á.); ybadilla@itcr.ac.cr (Y.B.-V.)
* Correspondence: rmoya@itcr.ac.cr; Tel.: +506-2550-9092

**Abstract:** The *Dipteryx* genus has a natural distribution throughout several tropical countries in Latin America. This taxon has several tree species, all recognized for their high-density wood. The objective of this research was to study the variation and genetic control of several wood properties, including bark, sapwood, heartwood, green density (GD), specific gravity (SG), moisture content in green condition (MC-G), and mechanical properties, in a *Dipteryx panamensis* provenance/progeny test at 8 years old. The results showed that bark varied little among families and provenances, but heartwood (diameter and percentage) showed high genetic variation. SG and MC-G showed significant variation between provenances and families, while GD showed little variation. Among the mechanical properties evaluated, the greatest genetic variation was observed in the MOR in bending and shear stress. Families from the Coope San Juan provenance registered the highest values in all wood properties investigated, and families from Puerto Viejo obtained the lowest. Family heritability and the coefficient of genetic variation exhibited high values in heartwood/sapwood and the MOR in bending ($h^2 > 0.9$ and CV > 20%) and lower values in SG, MC-G, compression stress, and shear stress. *D. panamensis* wood properties have a high potential to be improved through breeding programs.

**Keywords:** almendro; wood properties; breeding; native tree species



## 1. Introduction

Wood from *Dipteryx panamensis* (Pittier) Record and Mell trees from natural forests is included in Appendix III of CITES and this species is protected by Costa Rica's government [1,2]. However, this species has become popular in reforestation programs for timber production in Costa Rica [3,4]. Its natural distribution includes populations in the wet lowland forests from Nicaragua to Colombia [5,6]. Several studies reported its potential for commercial timber production as a medium growth rate tree species in plantations [7–10]. Planted trees of *D. panamensis* produce timber with suitable wood characteristics, such as high specific gravity and adequate mechanical properties for structural use and energy production [11]. *D. panamensis* is part of a select group of native tree species, recently defined as a strategic priority for research and development in the country [3]. In recent years the tree has received more attention from the industry for its favorable wood characteristics for structural usage [12].

*D. panamensis* has an adequate diameter growth rate in commercial plantations [8], rendering it as a suitable tree to undertake breeding programs [9,13]. A broad base seed collection from natural populations was followed in order to establish the first breeding program for *D. panamensis* in the region, as conventionally recommended [14]. Seedlings were obtained and established in repeated provenance/progeny trials in different environmental conditions. Breeding efforts were oriented to select the best adapted trees within the best families and the best provenances, based on superior growth performance, stem quality, and wood properties [9,10,13].

In this initial stage, provenances and families were gathered through open pollination collections from natural populations in order to capture the broadest genetic variation [13]. The first results showed that through adequate selection of the best individuals within the best families, up to 20% genetic progress in commercial volume can be achieved. Recent research with *D. panamensis* determined that the greatest genetic variability is found among individuals and among families within each provenance. However, there is little genetic variation among populations [9,13]. The studies suggest that a better strategy would be joining all provenances together into a single breeding population. In this manner, a higher genetic gain and enough improved seed volume supply can be obtained for local markets.

Despite the progress in breeding *D. panamensis*, it is still necessary to understand how the genotype by environment interactions (GxE) effect can produce a reduction in genetic gains [9]. Reforestation programs must be based on reliable seed sources, with the highest possible genetic quality and minimal risk in plantations [14]. Costa Rica is a land with an extraordinary diversity of climatic regions, which may promote important genotype by environment interactions (GxE) that may reduce genetic gains in breeding programs at a national scale [15,16]. The existence of this interaction could be of relevance and needs to be determined [7]. It is also important to mention that an effective breeding program, in addition to achieving an increase in genetic gain in growth and yield, must look at some of the essential wood properties for the industry [17].

Given the importance of *D. panamensis* in the production of wood for structural use in Costa Rica and in the region, the aim of this study was to (i) investigate the genetic control of key wood traits at the family level and within provenances and (ii) explore whether the improvement of solid wood traits is compatible with the energy focused programs in *D. panamensis*. Knowledge of the behavior of these traits will enhance and improve breeding efforts for this important native species in the region.

## 2. Materials and Methods

### 2.1. Sampling Area

The location of the provenance/progeny trial (8-year-old trees) was in Santa Clara, San Carlos, in the northern region in Costa Rica (10°21′ N and 84°30′ O) (Figure 1a). The life zone is classified as a very humid premontane forest transition to basal [18], with an average temperature and rainfall between 18 and 24 °C and 3800 and 4000 mm, respectively. The soils are reddish, acidic, and clay-rich Ultisols, deep and well drained [9], presenting average nutritional characteristics: 4.68 pH, 3.30 Ca, 1.57 Mg, 0.07 K, 31 Cu, 22 Mn, 105 Fe, 4.38 Zn, and 9.93 P [10].

### 2.2. Genetic Material and Trial Establishment

The genetic material for the provenance/progeny trials came from a collection of 17 families of *Dipteryx panamensis*, gathered from three native provenances in the northern region of Costa Rica: Coope San Juan at Cutris, San Carlos (CSJ); Crucitas at Pocosol, San Carlos (C); and Puerto Viejo, Sarapiquí (PV) (Figure 1a). These provenances are geographically separated by 50 to 70 km. From each of the provenances, open-pollinated seed was collected from 10 native mother trees (half-sib families), separated from each other by more than 500 m. More information about the trials is documented in León et al. [9].

The trial was designed and established in June 2010. Wood samples were obtained in April 2018, at 8 years old. The experimental design consisted of six complete random blocks [9]. Three pairs from each family were randomly distributed within each block (Figure 1b, with $n = 6$ as the experimental unit or plot). The trees were planted using a 3 m × 3 m spacing, at an initial density of 1111 trees/ha. Filler trees (indicated with R in Figure 1b) and two border rows around the entire trial were used. The site had no soil preparation nor control of soil acidity. Weed control was performed every 3 months until the age of 3 years. The first thinning was performed at 50% intensity at four years of age, consisting of removing the tree with the smallest diameter and worst stem quality from each pair within each block.

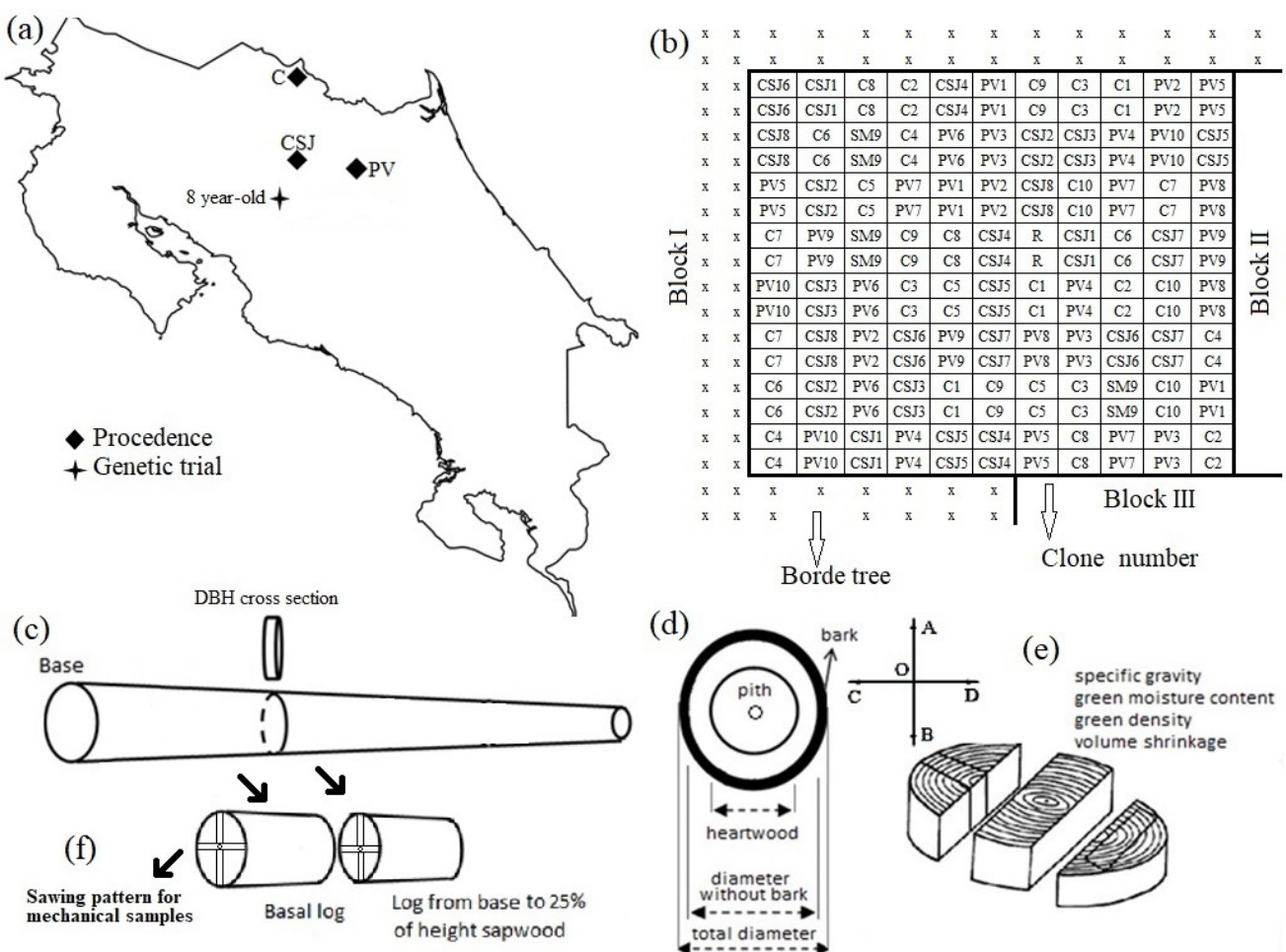

**Figure 1.** (**a**) Geographic location of the provenance/progeny trial of 8-year-old *Dipteryx panamensis*, northern region of Costa Rica: Coope San Juan (CSJ), Puerto Viejo (PV), and Crucitas (C) provenances. (**b**) Distribution of families within the block. (**c**) Sampled cross-section extraction from stem. (**d**) Parameters measured in cross-section and (**e**) cut pattern of cross-section for physical properties determination. (**f**) Sawing pattern for mechanical properties.

### 2.3. Sampling of Trees

Four individuals were selected from each of the 17 families present in the trial, totaling 68 individuals: 20 individuals from Coope San Juan provenance (CSJ), 20 individuals from Puerto Viejo provenance (PV), and 28 individuals from Crucitas provenance (C). The trees selected were codominant individuals, in order to leave standing the dominant trees and with diameters from 15 to 17 cm, corresponding to the average diameter at breast height (DBH) at that age. From each sampled tree, cross-section discs 3 cm thick were extracted at 1.30 m (DBH) along with two logs for mechanical properties (Figure 1c).

### 2.4. Morphological Variables

For each cross-section disc, tree diameter at DBH, bark thickness (BT), bark percentage (BP), sapwood thickness (SWT), sapwood total area percentage (SWP), heartwood diameter (HWD), and heartwood percentage (HWP) were determined. Two perpendicular lines were drawn crossing the center of each disc, one in the A–B direction and the other in the C–D direction (Figure 1b). Total diameter, diameter without bark, and HWD were measured in both directions on the disc (Figure 1d). The averages of the two transversal measurements were calculated to obtain both total diameter and HWD. BT and SWT were obtained as the difference between total diameter and diameter without bark (in the case of the BT) and HWT (in the case of sapwood). Diameters were calculated assuming a geometric circle.

The BP, SP, and HWP were calculated relative to the area of each cross-section section and the total area of the cross-sections.

### 2.5. Physical Properties

The physical properties determined were green density (GD), specific gravity (SG), moisture content in green condition (MC-G), tangential shrinking (TS), and radial shrinking (RS). The 3.0 cm wide wood pieces were cut at the DBH (Figure 1e) and they were divided at the pith, obtaining two samples from each one (Figure 1e). GD was calculated by the ratio of green weight/green volume, SG, TS, RS, and green MC-G according to ASTM D-143 and ASTM D-4442 procedures [19,20].

Wood color was determined in the heartwood area in subsamples A and B taken at DBH, at 12% (air dried) [16]. Wood color was measured using a portable colorimeter of a Hunter Lab Miniscan XE plus (Reston, VA, USA) and the CIEL*a*b* color system was utilized. The CIEL*a*b* color system estimates the value of three variables: coordinate L* for lightness, representing the position on the black–white axis (L* = 0 for black, L* = 100 for white); coordinate a* for the position on the red–green axis (positive values for red, negative values for green); and coordinate b* for the position on the yellow–blue axis (positive values for yellow, negative values for blue).

### 2.6. Mechanical Properties

The mechanical strengths determined were static bending (modulus of rupture and modulus of elasticity), lateral and axial Janka hardness, compression parallel to the grain, and shear strength parallel to the grain at 12% in moisture content conditions. The tests were performed following the ASTM D143-14 standards [19], and 18 specimens per family were prepared for each one of the mechanical tests.

Physical and mechanical properties were tested without division into sapwood and heartwood.

### 2.7. Statistical Analysis

A general statistical description (average and coefficient of variation) was applied by all wood properties. Before an analysis of variance was performed, data were checked for normality of distribution and homogeneity of variances. The ANOVA (analysis of variance) was performed to determine the family effect on wood properties. A means analysis was applied using the Tukey multiple range test ($p < 0.01$). SAS software was utilized for statistical analysis (Cary, NC, USA).

### 2.8. Genetic Analyses

Data were analyzed using model 1 from SELEGEN REML/BLUP software version 2008 [21]. The statistical model was

$$X_r = Z_r + W_p + \varepsilon \tag{1}$$

where $r$ is a vector of repetition effects (assumed as fixed) and added to the general mean, $p$ is a vector of plot effects assumed as random, and $\varepsilon$ is a vector of residuals (random). Capital letters represent incidence matrices for the mentioned effects and the significance level of the model was $p < 0.01$.

The following genetic parameters were estimated: individual heritability ($h^2a$) (Equation (2)), mean family heritability ($h^2mFam$) (Equation (3)), additive within family heritability ($h^2ad$) (Equation (4)), individual additive genetic variation coefficient ($CV_{gi}$) (Equation (5)), and genotypic variation coefficient among families ($CV_gFam$) (Equation (6)).

$$h^2a = V_{ad}/V_{plot} \tag{2}$$

$$h^2mFam = \frac{0.25 * V_{ad}}{0.25 * V_{ad} + \frac{V_{plot}}{6} + \frac{0.25 * V_{ad} + V_e}{18}} \tag{3}$$

$$h^2 ad = 0.75 * V_{ad} / (0.75 * V_{plot} + V_{ad}) \tag{4}$$

$$CV_{gi} = \frac{\sqrt{accuracy\ of\ family\ selection\ suming\ complete\ survival}}{General\ mean} * 100 \tag{5}$$

$$CV_g Fam = \frac{\sqrt{accuracy\ of\ family\ selection\ suming\ complete\ survival}}{General\ mean} * 100 \tag{6}$$

where 6 is the number of replications (r = 6) and 3 is the number of trees within each plot after thinning ($n$ = 3); thus, 18 is the total number of effective trees per family in the test. $V_{ad}$ is the additive variance, $V_{plot}$ is the variation among families within plots, and $V_e$ is the error variance.

## 3. Results

### 3.1. Morphological, Physical, and Mechanical Properties

The average values with their respective coefficients of variation for the provenances and families are presented in Table 1. DBH and BT ranged from 14.9 to 16.3 cm, and from 0.38 to 0.47, respectively, with no statistically significant differences between families or provenances. BP ranged from 9.4% to 11.9% and only the CSJ provenance showed statistically significant differences between families. HWD ranged from 3.5 to 7.2 cm and there were differences in families in the C and CSJ provenances. In the C provenance, the highest value was observed in the 9-family and the lowest value in the 10-family. In the CSJ provenance, the statistically highest values were observed in the 3-family and the 5-family. HWP ranged from 4.8 to 17.3% and differences between families were found in all provenances. In the C provenance, HWP was highest in the 4-, 5-, 6-, and 9-family, and the lowest percentage was in the 10-family. In the CSJ provenance, the highest HWP was obtained in the 3- and 5-family and the lowest in the 6-family, while in the PV provenance, the highest HWP value was observed in the 9-family and the rest of the families did not differ.

**Table 1.** Morphological properties variation at 8-years-old among native provenances and families of *Dipteryx panamensis* in Costa Rica.

| Prove-nance | Family | DBH (cm) | Bark Thickness (BT in cm) | Bark Percentage (BP in %) | Heartwood Diameter (HWD in cm) | Heartwood Percentage (HWP in %) | Sapwood Thickness (SWT in cm) | Sapwood Percentage (SWP in %) |
|---|---|---|---|---|---|---|---|---|
| C | 2 | 15.4 (7.0) A | 0.40 (12.5) A | 10.1 (6.1) A | 3.8 (11.1) BC | 6.1 (21.1) B | 5.4 (8.1) A | 83.8 (1.2) A |
| | 3 | 15.3 (3.1) A | 0.47 (8.0) A | 11.9 (8.0) A | 3.7 (17.5) BC | 6.1 (29.3) B | 5.3 (3.1) A | 82.1 (2.0) AB |
| | 4 | 14.9 (5.1) A | 0.46 (9.4) A | 11.9 (5.0) A | 4.1 (22.2) BC | 7.5 (32.9) AB | 4.9 (2.5) AB | 80.5 (3.5) AB |
| | 5 | 15.8 (3.6) A | 0.43 (21.9) A | 10.6 (19.3) A | 4.6 (13.0) BC | 8.7 (23.0) AB | 5.1 (5.4) A | 80.7 (2.4) AB |
| | 6 | 15.5 (6.0) A | 0.42 (3.0) A | 10.6 (4.2) A | 5.1 (15.8) B | 11.2 (31.4) A | 4.8 (12.1) AB | 78.2 (4.5) B |
| | 9 | 16.1 (4.4) A | 0.40 (8.8) A | 9.7 (12.5) A | 6.8 (7.1) A | 7.8 (12.7) AB | 4.3 (8.6) B | 72.5 (4.1) C |
| | 10 | 15.2 (6.8) A | 0.37 (10.2) A | 9.5 (14.8) A | 3.5 (14.0) C | 5.4 (14.9) C | 5.5 (5.4) A | 85.1 (1.1) A |
| CSJ | 1 | 15.6 (3.6) A | 0.42 (3.0) A | 10.5 (1.3) AB | 4.6 (13.8) B | 8.9 (27.3) B | 5.1 (7.7) AB | 80.6 (3.0) A |
| | 3 | 16.2 (4.1) A | 0.39 (8.0) A | 9.4 (6.6) B | 7.2 (19.2) A | 19.9 (33.9) A | 4.1 (13.0) B | 70.6 (9.1) B |
| | 4 | 15.3 (4.3) A | 0.40 (5.1) A | 10.2 (4.0) AB | 4.5 (12.8) B | 8.7 (20.4) B | 5.0 (5.4) AB | 81.1 (2.5) A |
| | 5 | 15.7 (4.5) A | 0.45 (12.0) A | 11.1 (10.1) A | 6.5 (9.7) A | 17.3 (22.9) A | 4.2 (12.8) B | 71.6 (6.6) B |
| | 6 | 15.9 (4.2) A | 0.40 (12.5) A | 9.8 (10.4) AB | 3.4 (19.8) B | 4.8 (40.0) C | 5.9 (7.6) A | 85.5 (2.4) A |
| PV | 2 | 15.7 (7.5) A | 0.41 (18.2) A | 10.1 (11.6) A | 4.1 (11.0) A | 6.9 (16.8) B | 5.4 (7.6) A | 82.9 (1.4) A |
| | 3 | 16.1 (3.4) A | 0.44 (9.9) A | 10.5 (6.7) A | 4.7 (13.2) A | 8.6 (24.5) B | 5.3 (6.3) AB | 80.8 (2.4) A |
| | 8 | 15.7 (5.5) A | 0.44 (15.5) A | 11.0 (17.4) A | 5.6 (12.6) A | 12.6 (18.5) A | 4.6 (5.7) B | 76.4 (2.4) A |
| | 9 | 15.4 (8.6) A | 0.38 (12.2) A | 9.5 (4.1) A | 4.5 (23.0) A | 8.5 (34.7) B | 5.1 (8.1) AB | 82.0 (4.0) A |
| | 10 | 16.3 (3.1) A | 0.41 (14.4) A | 9.9 (11.8) A | 4.2 (5.5) A | 6.6 (4.9) B | 5.6 (1.6) A | 83.5 (1.7) A |
| General average | | 15.7 (5.2) | 0.42 (12.4) | 10.4 (11.5) | 4.8 (26.6) | 9.8 (52.0) | 5.0 (11.7) | 79.9 (6.4) |

Note: Different letters mean statistical differences at 99% among families per provenance. The values in parentheses present the coefficient of variation.

SWT ranged from 4.1 to 5.9 cm and all provenances showed differences among their families. In the C provenance, the 9-family registered the lowest value in relation to the rest of the families. In the CSJ provenance, the lowest SWT value was observed in the 3-

and 5-family. In the PV provenance, the lowest value was in the 8-family. SWP varied between 70.6 and 85.5 and the differences between families were found in the C and CSJ provenances. In the C provenance, the 2- and 10-family presented the highest SWP values and the lowest SWP value was in the 9-family, while in the CSJ provenance, the families with the highest percentages were the 6-, 4-, and 1-family (Table 1).

### 3.2. Physical Properties

In the statistical analysis of physical properties by provenance, it was determined that there were no significant statistical differences among the three provenances studied for any of the variables (Figure 2a–e).

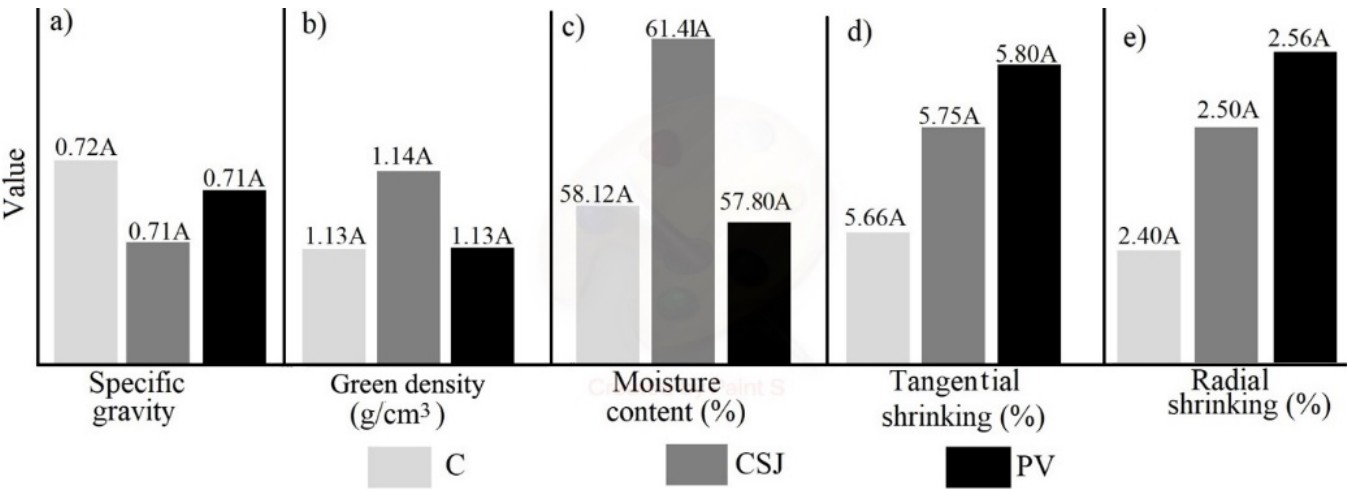

**Figure 2.** Physical properties at 8-years-old among native provenances of *Dipteryx panamensis* in Costa Rica: specific gravity-SG (**a**), green density-GD (**b**), green moisture content-MC-G (**c**), tangential shrinking (**d**) and radial shrinking (**e**). Note: Different letters indicate statistically significant differences at 99% among provenances.

Effects of the physical properties by family and provenance are presented in Table 2. The table shows that there are significant differences between families within one region in terms of SG, G-MC, TS, and the a* color parameter of heartwood. The SG ranged from 0.64 to 0.77 for all provenances. The highest value was observed in the 4-family and the lowest value was found in the 10-family. In the CSJ provenance, the statistically higher value was obtained in the 4-family and the lowest value was in the 5-family (Table 2). GD ranged between 1.1 and 1.2 g/cm$^3$ and no statistically significant differences were observed among families of the three provenances (Table 2). MC-G varied between 52.8% and 74.7% and only in the CSJ provenance were there differences among families. The 5-family registered the highest value and the 4-family had the statistically lowest value (Table 2). TS varied between 5.3% and 6.5% and only the CSJ provenance showed differences between families. The highest percentage was presented by the 1-family, while the 3-family registered the statistically lowest percentage (Table 2). RS varied between 2.2% and 2.7% and there were no statistically significant differences between families of the provenances. In the color parameters, only the a* color parameter showed differences between families, while the L* and b* color parameters did not show differences among families (Table 2). The a* color parameter varied between 6.0 and 9.6. In the C provenance, the highest value was observed in the 9-family, while the lowest was found in the 3-family. In the CSJ provenance, families 6 and 3 showed statistically significant differences between them (Table 2).

### 3.3. Mechanical Properties

The statistical analysis of the mechanical properties showed statistically significant differences between the three provenances evaluated in the MOR and MOE of the flexion test and lateral Janka hardness (Figure 3a,b,f), but for other mechanical properties no

statistically significant differences were found. The MOR values from the bending test were lower for the PV provenance than for the other two provenances (Figure 3a), while for the MOE, the C provenance was superior to CSJ and PV (Figure 3b). The compressive strength in shear and axial Janka hardness did not differ between provenances (Figure 3c–e), while in lateral compression Janka hardness, the C provenance was superior to CSJ and PV (Figure 3f).

**Table 2.** Physical properties at 8-years-old among native provenances and families of *Dipteryx panamensis* in Costa Rica.

| Prove-nance | Family | Specific Gravity (SG) | Green Density (GD in g/cm³) | Green Moisture Content (MC-G in %) | Tangential Shrinking (TS in %) | Radial Shrinking (RS in %) | Color L* | a* | b* |
|---|---|---|---|---|---|---|---|---|---|
| C | 2 | 0.72 (4.4) [B] | 1.1 (2.4) [A] | 57.4 (11.4) [A] | 5.6 (9.6) [A] | 2.5 (15.0) [A] | 58.1 (3.4) [A] | 8.1 (19.3) [AB] | 28.1 (9.7) [A] |
| | 3 | 0.72 (2.8) [B] | 1.1 (2.2) [A] | 59.8 (3.7) [A] | 5.4 (11.3) [A] | 2.2 (13.6) [A] | 62.7 (6.5) [A] | 6.8 (12.1) [B] | 25.7 (11.5) [A] |
| | 4 | 0.77 (3.2) [A] | 1.1 (6.9) [A] | 55.9 (7.4) [A] | 6.1 (12.6) [A] | 2.3 (26.4) [A] | 61.3 (7.5) [A] | 7.2 (25.7) [AB] | 26.6 (11.6) [A] |
| | 5 | 0.69 (7.2) [B] | 1.1 (5.2) [A] | 61.3 (6.9) [A] | 5.8 (12.3) [A] | 2.4 (10.5) [A] | 62.1 (6.7) [A] | 6.9 (12.3) [AB] | 28.1 (10.2) [A] |
| | 6 | 0.72 (3.4) [B] | 1.1 (2.8) [A] | 58.1 (10.1) [A] | 5.8 (7.6) [A] | 2.7 (20.6) [A] | 60.9 (6.2) [A] | 8.6 (21.8) [AB] | 27.6 (10.0) [A] |
| | 9 | 0.70 (6.0) [B] | 1.1 (4.8) [A] | 60.4 (12.5) [A] | 5.3 (1.5) [A] | 2.2 (15.8) [A] | 58.0 (4.9) [A] | 9.6 (33.8) [A] | 28.3 (17.1) [A] |
| | 10 | 0.73 (2.1) [AB] | 1.1 (3.9) [A] | 53.9 (8.6) [A] | 5.6 (10.0) [A] | 2.4 (14.6) [A] | 63.2 (4.6) [A] | 7.0 (13.9) [AB] | 27.5 (5.4) [A] |
| CSJ | 1 | 0.72 (3.2) [BC] | 1.1 (3.4) [A] | 59.1 (8.2) [BC] | 6.5 (8.5) [A] | 2.4 (18.2) [A] | 67.2 (3.4) [A] | 6.1 (15.6) [AB] | 27.6 (6.0) [A] |
| | 3 | 0.69 (4.6) [C] | 1.1 (2.2) [A] | 64.9 (8.8) [B] | 5.4 (2.3) [B] | 2.5 (5.1) [A] | 63.8 (7.3) [A] | 6.0 (32.9) [B] | 28.0 (7.9) [A] |
| | 4 | 0.76 (1.2) [A] | 1.2 (1.1) [A] | 52.8 (3.5) [C] | 5.7 (7.7) [AB] | 2.4 (15.9) [A] | 63.2 (9.6) [A] | 6.9 (14.4) [AB] | 29.2 (4.3) [A] |
| | 5 | 0.64 (3.7) [D] | 1.1 (2.8) [A] | 74.7 (9.9) [A] | 5.6 (8.9) [B] | 2.5 (13.5) [A] | 64.6 (4.8) [A] | 7.5 (19.1) [AB] | 30.9 (9.8) [A] |
| | 6 | 0.74 (3.7) [AB] | 1.2 (2.6) [A] | 55.5 (11.5) [C] | 5.6 (2.3) [B] | 2.7 (10.9) [A] | 61.6 (6.9) [A] | 7.9 (11.9) [A] | 27.6 (15.9) [A] |
| PV | 2 | 0.70 (1.9) [A] | 1.1 (4.3) [A] | 57.5 (10.8) [A] | 5.5 (3.4) [A] | 2.2 (12.1) [A] | 65.2 (6.9) [A] | 6.6 (21.7) [A] | 26.8 (12.9) [A] |
| | 3 | 0.71 (1.9) [AB] | 1.1 (4.6) [A] | 54.9 (9.7) [A] | 5.8 (7.4) [A] | 2.7 (20.6) [A] | 61.4 (4.1) [A] | 6.8 (22.4) [A] | 28.1 (12.3) [A] |
| | 8 | 0.71 (5.2) [AB] | 1.1 (5.4) [A] | 61.0 (8.9) [A] | 5.8 (4.5) [A] | 2.5 (12.7) [A] | 63.9 (5.6) [A] | 6.7 (11.1) [A] | 30.7 (8.8) [A] |
| | 9 | 0.72 (2.8) [AB] | 1.1 (6.0) [A] | 61.3 (6.2) [A] | 5.6 (5.4) [A] | 2.7 (21.5) [A] | 65.5 (4.5) [A] | 6.7 (18.2) [A] | 28.6 (8.8) [A] |
| | 10 | 0.74 (3.4) [B] | 1.1 (4.0) [A] | 54.2 (8.6) [A] | 6.3 (10.7) [A] | 2.7 (12.2) [A] | 64.9 (5.4) [A] | 7.2 (31.0) [A] | 29.2 (9.8) [A] |
| General average | | 0.72 (5.4) | 1.17 (3.9) | 59.9 (12.1) | 5.7 (9.2) | 2.5 (15.9) | 62.8 (18.2) | 7.2 (29.1) | 28.1 (20.1) |

Note: Different letters mean statistical differences at 99% among families per provenance. The values in parentheses present the coefficient of variation.

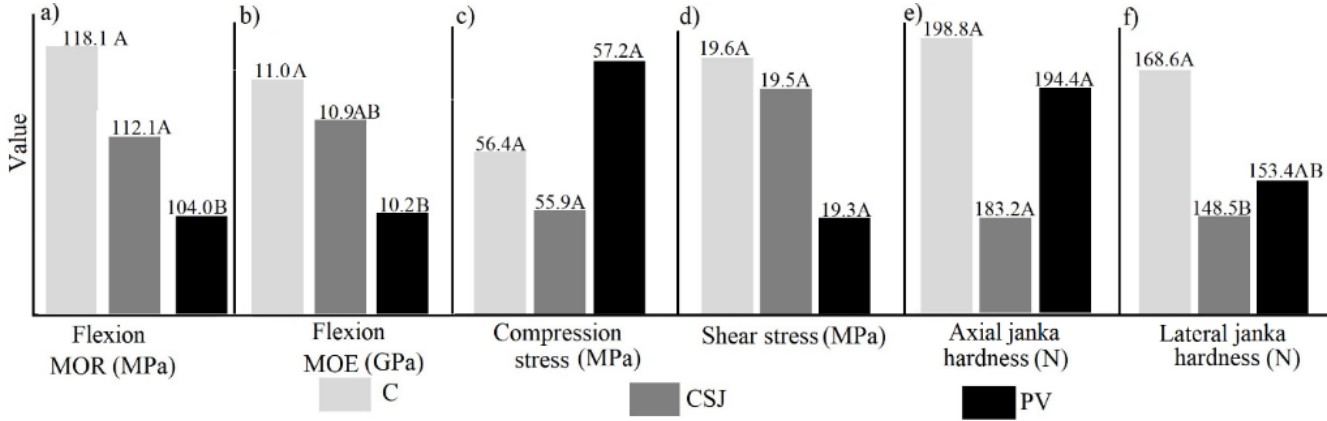

**Figure 3.** Mechanical properties at 8-years-old among native provenances of *Dipteryx panamensis* in Costa Rica: Module of rupture-MOR in flexure test (**a**), module elasticity-MOE in flexure test (**b**), compression stress (**c**), shear stress (**d**), axial janka hardness (**e**) and lateral janka hardness (**f**). Note: Different letters indicate statistically significant differences at 99% among provenances.

Differences among families of the same provenance also reflected a high variation (Table 3). In the MOR in the bending test, the 2- and 3-family of the C provenance presented the lowest values. In the CSJ provenance, the lowest value was observed in the 10-family and the 1-family had the highest value. In the PV provenance, the 8-family had the lowest MOR value (Table 3). The MOE in the bending test showed no statistically significant

differences between families within each provenance (Table 3). Regarding compression stress, the families showed no differences between the C and CSJ provenances. In the PV provenance, the 10-family registered the statistically lowest value (Table 3). In terms of the shear test, the 2-, 3-, 8-, and 10-family of the PV provenance presented the statistically lowest values, while in the CSJ provenance, differences were presented between the 1-family and the 6-family. In the PV provenance, the 8-family recorded the lowest value. The axial and lateral Janka hardness in the C and PV provenances did not show statistically significant differences between families. In the CSJ provenance, there were differences between families, where the 4-family presented the highest hardness values (Table 3).

**Table 3.** Mechanical properties at 8-years-old among native provenances and families of *Dipteryx panamensis* in Costa Rica.

| Provenance | Family | Bending Test | | Compression Stress (MPa) | Shear Stress (MPa) | Janka Hardness (N) | |
|---|---|---|---|---|---|---|---|
| | | MOR (MPa) | MOE (MPa) | | | Axial | Lateral |
| C | 2 | 99.6 (11.5) [B] | 1170.2 (11.6) [A] | 56.8 (9.0) [A] | 18.4 (7.3) [C] | 194.5 (15.8) [A] | 157.3 (14.4) [A] |
| | 3 | 88.0 (16.3) [B] | 988.7 (15.6) [A] | 54.1 (8.4) [A] | 18.7 (7.2) [BC] | 195.4 (21.2) [A] | 160.4 (15.5) [A] |
| | 4 | 118.4 (17.6) [A] | 1060.9 (14.5) [A] | 54.1 (9.9) [A] | 21.5 (9.6) [A] | 219.2 (34.2) [A] | 220.6 (93.9) [A] |
| | 5 | 122.8 (12.8) [A] | 1052.5 (39.6) [A] | 56.3 (15.4) [A] | 18.5 (10.5) [AB] | 204.9 (20.7) [A] | 156.6 (18.5) [A] |
| | 6 | 128.8 (10.4) [A] | 1117.8 (18.0) [A] | 60.5 (11.1) [A] | 20.9 (7.4) [A] | 179.2 (12.8) [A] | 157.2 (13.6) [A] |
| | 9 | 132.4 (14.3) [A] | 1057.0 (19.2) [A] | 56.2 (12.1) [A] | 18.8 (8.2) [C] | 200.8 (19.5) [A] | 160.4 (16.2) [A] |
| | 10 | 133.5 (10.1) [A] | 1138.5 (13.8) [A] | 56.9 (9.2) [A] | 19.9 (5.3) [ABC] | 197.2 (13.6) [A] | 167.7 (26.2) [A] |
| CSJ | 1 | 121.9 (13.4) [A] | 1137.5 (27.0) [A] | 53.7 (9.7) [A] | 18.5 (8.8) [B] | 175.0 (16.6) [A] | 143.9 (23.3) [B] |
| | 3 | 110.9 (19.2) [AB] | 1021.7 (22.2) [A] | 55.5 (12.7) [A] | 20.0 (14.9) [AB] | 186.3 (19.5) [AB] | 144.4 (25.5) [B] |
| | 4 | 113.8 (9.6) [AB] | 1059.6 (31.6) [A] | 57.9 (6.0) [A] | 19.8 (6.6) [AB] | 209.9 (22.2) [B] | 178.5 (20.2) [A] |
| | 5 | 104.5 (12.00) [AB] | 946.0 (31.1) [A] | 54.2 (8.1) [A] | 18.9 (7.3) [AB] | 175.5 (16.9) [AB] | 129.1 (16.3) [B] |
| | 6 | 93.4 (14.3) [B] | 1149.9 (23.9) [A] | 58.2 (8.9) [A] | 20.4 (6.6) [A] | 169.2 (12.8) [A] | 146.4 (16.0) [B] |
| PV | 2 | 106.9 (12.1) [A] | 1011.9 (14.2) [A] | 57.1 (8.8) [AB] | 20.4 (7.5) [A] | 188.4 (17.8) [A] | 145.3 (17.2) [A] |
| | 3 | 100.3 (11.1) [A] | 1049.8 (13.5) [A] | 54.9 (12.5) [B] | 18.8 (7.3) [B] | 202.6 (25.0) [A] | 162.1 (21.9) [A] |
| | 8 | 92.0 (12.3) [B] | 990.1 (24.9) [A] | 54.1 (9.4) [B] | 16.6 (6.6) [C] | 186.3 (20.8) [A] | 154.4 (31.6) [A] |
| | 9 | 107.5 (15.0) [A] | 1011.4 (23.6) [A] | 56.5 (13.7) [B] | 20.2 (7.5) [AB] | 198.6 (14.5) [A] | 148.9 (24.4) [A] |
| | 10 | 107.7 (11.9) [A] | 1049.8 (17.2) [A] | 63.2 (11.6) [A] | 20.7 (6.7) [A] | 196.2 (18.3) [A] | 156.3 (20.4) [A] |
| General average | | 112.4 (10.7) | 1070.2 (17.2) | 56.5 (11.1) | 19.5 (10.1) | 192.9 (20.7) | 158.2 (38.1) |

Note: Different letters mean statistical differences at 99% among families within provenances. The values in parentheses present the coefficient of variation.

### 3.4. Genetic Analysis

The data analysis determined significant differences between families in the variables HWD, SWT, HWP, SWP, SG, MC-G, TS, color parameters L* and a*, MOR in the bending test, compressive stress, shear stress, axial Janka hardness, and lateral Janka hardness (Table 4). With respect to the analysis between provenances, significant variation between families was determined for the variables HWD, SWT, HWP, SWP, MC-G, L* and a* color parameters, MOR, and MOE in bending and axial Janka hardness (Table 4). Regarding the degree of genetic control, high individual heritability values were determined for bark thickness and percentage, radial shrinkage, L* and a* color parameters, compressive stress, and both Janka variables. In terms of family heritability, significant values were found in the variables bark thickness and percentage, HWD, SWT, HWP, SWP, SG, MC-G, MOR and MOE in bending, and both Janka variables.

In addition to these high heritability values for most variables, high values of $CV_{gi}$ (>10%) were found for most of them, especially relevant in HWD and HWP, which recorded the $h^2mFam$ values as well.

**Table 4.** Genetic parameters of *Dipteryx panamensis* wood properties in native provenances and families from Costa Rica.

| Variable | F-Value for Family | F-Value for Provenance | $h^2a$ | Family | | |
| --- | --- | --- | --- | --- | --- | --- |
| | | | | $h^2mFam$ | $Cv_{gi}$ | $CV_gFam$ |
| DBH | 0.6 | 1.4 | 0.01 | 0.01 | 0.52 | 0.26 |
| Bark thickness (BT) | 1.31 | 0.19 | 0.27 | 0.22 | 6.44 | 3.22 |
| Bark percentage (BP) | 1.95 | 1.18 | 0.80 | 0.50 | 10.44 | 5.22 |
| Heartwood diameter (HWD) | 7.75 * | 6.6 * | - | 0.91 | 44.94 | 22.47 |
| Heartwood percentage (HWP) | 8.2 * | 9.07 * | - | 0.92 | 89.95 | 44.97 |
| Sapwood thickness (SWT) | 6.11 * | 4.59 * | - | 0.87 | 18.60 | 9.30 |
| Sapwood percentage (SWP) | 8.72 * | 7.46 * | - | 0.92 | 10.96 | 5.48 |
| Specific gravity (SG) | 14.95 * | 1.29 | - | 0.78 | 7.26 | 3.63 |
| Green density (GD) | 0.71 | 0.67 | - | 0.00 | 0.24 | 0.12 |
| Green moisture content (MC-G) | 9.93 * | 5.88 * | - | 0.73 | 15.01 | 7.51 |
| Tangential shrinking (TS) | 2.36 * | 0.54 | - | 0.03 | 2.71 | 1.36 |
| Radial shrinking (RS) | 0.73 | 1.08 | 0.65 | 0.44 | 7.52 | 3.76 |
| L* color parameter | 2.38 * | 11.18 * | 0.65 | 0.51 | 5.58 | 2.79 |
| a* color parameter | 2.9 * | 5.47 * | 0.30 | 0.28 | 13.37 | 6.68 |
| b* color parameter | 1.64 | 2.84 | 0.016 | 0.02 | 1.40 | 0.70 |
| MOR in bending | 11.82 * | 21.84 * | - | 0.94 | 24.18 | 12.09 |
| MOE in bending | 1.57 | 3.68 * | 0.14 | 0.35 | 6.57 | 3.29 |
| Compression stress | 3.38 * | 0.88 | 0.39 | 0.62 | 6.97 | 3.48 |
| Shear stress | 11.94 * | 0.33 | - | 0.89 | 11.75 | 5,87 |
| Janka hardness axial | 2.65 * | 3.57 * | 0.18 | 0.41 | 8.90 | 4.45 |
| Janka hardness lateral | 2.34 * | 2.89 | 0.16 | 0.39 | 15.38 | 7.69 |

Note: * Statistical significance at 95%. The absence of $h^2a$ values is due to the low number of samples available.

## 4. Discussion

### 4.1. Morphological, Physical, and Mechanical Properties

The values of BP, HWP, and SWP found for the different families (Table 1) are lower than those reported by Tenorio et al. [22] for *D. panamensis* trees. The values found for physical properties (SG, MC-G, GD, TS, and RS) and the different mechanical properties studied (MOR and MOE in bending test, compression stress, shear stress, and Janka hardness) for the different origins (Figure 3) and the different families (Table 2) of 8-year-old *D. panamensis* present some differences from those reported by Tenorio et al. [22] for *D. panamensis* trees from 13-year-old plantations. For example, these authors report higher values of SG, MC-G, GD, TS, and RS than those found in this study (Table 2), resulting in higher mechanical property values in the wood of Tenorio et al. [22]. On the other hand, the highest values of SG, GD, RT, and TS were reported for wood from trees growing in a natural forest [23], values higher than those obtained in this study for the different provenances (Figure 2) and the different families (Table 2).

The difference that occurred between the values of general properties (Table 1), physical properties (Figure 2, Table 2), and mechanical properties (Figure 3, Table 3) and those reported by Tenorio et al. [22] and Blanco et al. [23] is explained by the difference in tree age and maturity. In the present study, 8-year-old trees were sampled, which are juvenile wood; in the study conducted by Tenorio et al. [22], the trees tested were 13 years old, while in the study of Blanco et al. [23] the trees were from a natural forest, which indicates that they tested mature trees. When the trees (their trunks) achieve managed dimensions, i.e., dimensional usage in the wood industry, heartwood will dominate in the volume of the trunks.

Although it is observed that the general, physical, and mechanical properties of the present study of provenances and families are lower than those reported by other studies, the wood produced at this age reaches high values of SG and low values in shrinkages, so it can have structural use despite its juvenile state, attributed to adequate mechanical properties compared to other plantation wood grown in fast growing conditions in Costa Rica [4,22]. Another property of wood from these origins and families that favors industrial

processes in wood drying is the low MC-G value, since it decreases the drying time and increases the quality of the wood [24].

*4.2. Genetic Analysis*

Studies relating the effect of tree genetic origin to variations in wood properties are frequently reported [25,26]. However, for tropical hardwood species in the American region, the genetic effects on wood properties are limited to a few species [27–30]. The genus *Dipteryx* lacks information on genetic effect variation in wood, except for these same provenances and families, which have recently reported family effects on variation in wood properties along the tree height [31].

Heartwood/sapwood, bark, and pith tissue within tree formation are usually affected by the environment, but are under a strong genetic control as well [17]. This behavior was evident in the present study; specifically, there was a significant variation in these tissues between provenances (Figure 2), as well as between families for the same provenance (Table 2). These results are confirmed by the values of heritability and the genetic coefficient of variation (Table 4), where, with the exception of bark (thickness and percentage), provenance and family were significant. Individual ($h^2a$) and family ($h^2mFam$) heritability presented high values ($h^2 > 0.90$) for some wood properties. Morphological properties and heartwood parameters showed the highest $h^2mFam$ values (Table 4).

In relation to wood properties, significant differences between provenances occurred in mechanical and physical properties, although the differences were not evident (Figures 2 and 3). Specifically, there were significant differences between the PV provenance with CSJ and between PV with C for the MOR and MOE in bending, this parameter being lower in the PV provenance (Figure 3a,b). Thus, these results for mechanical properties suggest an important genetic effect attributable to the provenance [17]. The specific study of the physical and mechanical properties between families within each provenance shows the statistically significant differences between the properties investigated (Tables 2 and 3). This, together with the family heritability values ($h^2mFam$), again shows the potential for genetic improvement of these wood properties [32]. It is also important to mention that the coefficient of individual and family genetic variation complements the analysis of the genetic improvement potential of the investigated properties. In this study, the same properties with high values for heritability also showed high values for this genetic parameter.

If, for example, SG is considered as the most important physical property due to its influence on other wood properties [33], the 2-, 4-, and 10-family of the C provenance; 4- and 6-family of the CSJ provenance; and 2-, 3-, 8-, and 10-family of the PV provenance (Table 2), registered the highest potential for establishing a genetic improvement program for the physical and mechanical properties (Table 3).

While the environment affects almost all morphological, physical, and mechanical wood properties, some of them are also under strong genetic control, as has been frequently reported [17,34]. The environmental and soil conditions of each provenance have been correlated with wood properties in many species [17,34]. This fact is important, since heartwood, SG, and mechanical properties are essential for the commercialization of this hardwood species due to its aesthetic quality, durability, and resistance [22]. These results show the potential for the improvement of these wood properties in *Dipteryx*, based on the selection of superior individuals within the best families and provenances, as is customary in provenance/progeny trials with forest species [15]. However, this must be in congruence with the growth parameters of the trees. León et al. [9], in a study of these same populations, show that the selection of individuals and provenances using tree growth parameters is a viable strategy for the improvement of *Dipteryx panamensis* for sawlog production. These results, together with those recorded in this research for wood properties, considering provenances and families, serve as an important input for further genetic and silvicultural improvement [9].

### 4.3. Heritability of Wood Properties

The $h^2a$ and $h^2mFam$, as well as the genetic coefficients of variation ($CV_{gi}$ and $CV_gFam$) for bark and a good number of wood properties, showed moderate to very high values. These results allow us to infer that from relatively early ages it is possible to begin with the selection process. However, this should be taken with caution, since genetic control increases with age [35,36]. In this regard, it is necessary to continue with measurements over time of this type of genetic assay, with the purpose of determining the optimal age of selection.

Martínez et al. [37] and León et al. [9] indicate a high potential for genetic improvement in growth and stem quality in *D. panamensis*. High heritability values of up to 72% were recorded for growth traits and over 39% for qualitative traits such as apical dominance and the absence of thick branches. According to the genetic ranking of commercial volume presented by Martínez et al. [37], the 3-family of the PV provenance is one of the most productive. However, in terms of wood properties (SG and MOE in the bending test), this same family recorded lower values than the other materials evaluated (Figure 3a,b). Therefore, the data suggest that a greater diameter development could decrease some properties, such as SG and mechanical properties [34]. Authors should discuss the results and how they can be interpreted from the perspective of previous studies and of the working hypotheses. The findings and their implications should be discussed in the broadest context possible. Future research directions may also be highlighted.

### 5. Conclusions

The results presented in this study for *D. panamensis* should be considered as a first effort on breeding for wood properties. Future research is needed to validate and contrast these findings, with a larger number of families and provenances, and older age records.

Differences in morphological characteristics are few, except for heartwood-related parameters. Families from the Coope San Juan (CSJ) provenance recorded the highest values for the wood properties investigated. In contrast, families from the Puerto Viejo (PV) provenance had the lowest values for almost all properties. Therefore, this provenance should be considered more carefully in terms of breeding.

Individual and family heritability values recorded in this research were high for heartwood/sapwood and the MOR in bending, while values were moderate to high in SG, MC-G, compression stress, and shear stress. Thus, these wood properties of *Dipteryx panamensis* have potential for breeding favorable wood properties.

Finally, breeding efforts should concentrate on increasing the collection of provenances and native families in order to increase the genetic base of the current program, since selection based on commercial volume is not necessarily associated with the selection of the best individuals with respect to wood properties.

**Author Contributions:** Conceptualization, D.R.-P., O.M., Y.B.-V. and R.M.; methodology, D.R.-P., O.M., Y.B.-V. and R.M.; validation, D.R.-P., J.G.-Á. and R.M.; formal analysis, D.R.-P., Y.B.-V. and O.M.; investigation, D.R.-P. and R.M.; resources, R.M.; writing—original draft preparation, D.R.-P., O.M., J.G.-Á. and R.M.; writing—review and editing, D.R.-P., J.G.-Á. and R.M. SPS availability, Y.B.-V. All authors have read and agreed to the published version of the manuscript.

**Funding:** This research received no external funding.

**Data Availability Statement:** Not applicable.

**Acknowledgments:** The authors wish to thank the Division of Research and Extension at the Instituto Tecnológico de Costa Rica (ITCR) for its economic support, as well as GENFORES, the International Tree Improvement Cooperative, for the utilization of its genetic trials.

**Conflicts of Interest:** The authors declare no conflict of interest.

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
