# Peer review of "Variation and Genetic Control of the Heartwood, Sapwood, Bark, Wood Color Parameter, and Physical and Mechanical Properties of Dipteryx panamensis in Costa Rica"

_forests, doi:10.3390/f13010106_

Round 1
Reviewer 1 Report
The aim of the article is clearly stated: (i) investigate the genetic control of key wood traits at both families within provenance level. (ii) Explore whether the improvement of solid wood traits is compatible with the energy focused programs. The research is reach on data, whose are correctly statistically evaluated. This is not obvious in many articles, even published in “Forests”. Based on reference literature, there are few works about Dipteryx Panamensis from Costa Rica in comparison to e.g. Fagus sylvatica, Norway spruce… from over the world. It may be interesting to readers of Forests. The paper is not well written as I indicate in my review. Especially, the methods of research must be clearly described for beginners. The article is written for specialists in the field. The conclusions: “Differences in morphological characteristics are few, except for heartwood-related parameters.
Families from CoopeSanJuan (CSJ) provenance recorded the highest values for the wood properties investigated. In contrast, families from the Puerto Viejo (PV) provenance had the lowest values for almost all properties. Therefore, this provenance should be considered more carefully in terms of breeding” support aim (i). “The results presented in this study for D. panamensis should be considered as a first effort on breeding for wood properties, future research is needed” supports aim (ii). Nowadays, there is fight between researches whose are for human genetic control of products and researches whose supports the natural principles of genetics. This article is aimed to promote both streams. So, that the special for this article.
Line 25 and throughout the whole text: replace “flexure” to “bending”;
Line 108: square meter;
Line 177-182: Explain the symbols on the right sides of the formulas.
Line 253: Janka hardness with capital “J”;
Line 281 (Table 3.): Is one decimal place of MOE in MPa omitted? Or, explain in Methods the position of supports in bending!
Author Response
Review 1
Line 25 and throughout the whole text: replace “flexure” to “bending”;
Answer: word replaced in the test
Line 108: square meter;
Answer: is 3m x 3m spacing
Line 177-182: Explain the symbols on the right sides of the formulas.
Answer: we added in the test: “Vad: is the additive variance, Vplot: is the variation among families within plots, and Ve is the error variance.” (L184-L194)
Line 253: Janka hardness with capital “J”;
Answer: this observation was corrected
Line 281 (Table 3.): Is one decimal place of MOE in MPa omitted? Or, explain in Methods the position of supports in bending!
Answer: This observation was not considered. The presentation is confused.

Reviewer 2 Report
Dear Authors,
The manuscript is prepared very reliably and carefully.
I have only one significant substantial comment.
Chapter 2 Materials and Methods – Physical and mechanical properties (lines 134-154)
Many species of trees show clear differences in wood moisture content. I mean differences between the moisture content of sapwood and heartwood in living trees.
After cutting trees and supplying wood to the same level of moisture content, sapwood usually shows significantly different physical and mechanical properties compared to heartwood. In particular, this applies to compressive strength and hardness (lignin content is important here). This is crucial when testing the color. The wood of the genus Dipterix usually has a strongly colored heartwood.
This requires comment in methodology.
It should be clearly defined that physical and mechanical properties were tested without division into sapwood and heartwood, or maybe only sapwood was tested?
The above comment is crucial when discussing the results too – Chapter 4 (lines 319-332).
When comparing to older trees (mature wood), it should be taken into account that the studied 8-year-old trunks are 100% juvenile wood.
I hope that research will continue in the following years when the trees (their trunks) achieve managed dimensions, i.e. dimensional usage in the wood industry, and in the volume of the trunks heartwood will dominate.
Yours sincerely
Reviewer
Author Response
Review 2
- Chapter 2 Materials and Methods – Physical and mechanical properties (lines 134-154). Many species of trees show clear differences in wood moisture content. I mean differences between the moisture content of sapwood and heartwood in living trees.After cutting trees and supplying wood to the same level of moisture content, sapwood usually shows significantly different physical and mechanical properties compared to heartwood. In particular, this applies to compressive strength and hardness (lignin content is important here). This is crucial when testing the color. The wood of the genus Dipterix usually has a strongly colored heartwood. This requires comment in methodology.
Answer: The authors disagree this observation. However, Dipterix genius presents a strongly colored heartwood, the trees from fast-growth forest plantation for this species and many tropical species, are characterized for low heartwood content. The heartwood proportions varied from 6 to 20% (see table 1). Then the differences between sapwood and heartwood is limited to determine.
- It should be clearly defined that physical and mechanical properties were tested without division into sapwood and heartwood, or maybe only sapwood was tested?
Answer: comment added to the methodology
- When comparing to older trees (mature wood), it should be taken into account that the studied 8-year-old trunks are 100% juvenile wood.
Answer: observation added to the text
- I hope that research will continue in the following years when the trees (their trunks) achieve managed dimensions, i.e. dimensional usage in the wood industry, and in the volume of the trunks heartwood will dominate.
Answer: This observation was considered (L338-339)